# Transrenal *Mycobacterium tuberculosis* DNA in pulmonary tuberculosis patients during the first 14 days of treatment

Irina Kontsevaya,[1,2,3,4] Jan Heyckendorf,[5] Frauke Koops,[1,2] Doris Hillemann,[6] Torsten Goldmann,[7,8] Caryn M. Upton,[9] Veronique De Jager,[9] Andreas Diacon,[9] Christoph Lange[1,2,3,10]

**ABSTRACT**   We assessed the performance of a novel real-time PCR-based transrenal DNA (trDNA) assay for the specific detection of *Mycobacterium tuberculosis* as a candidate marker of the early anti-tuberculosis therapy response. The study was performed on 288 urine samples from 72 tuberculosis patients collected at baseline and days 3, 7, and 14 of treatment with amoxicillin-clavulanic acid alone or in combination with meropenem, ertapenem, optimized-dose rifampicin, or standard treatment control in South Africa. trDNA was detected in one-third of the samples. The highest proportion of positive PCR results (cycle threshold < 36) was observed on days 3 and 7, reflecting the point in time when maximum bacterial killing and disintegration are expected. When analyzed by study arms, the trend was observed in groups treated with active antibiotics affecting cell wall integrity (meropenem, control) but not in inactive drugs (ertapenem, amoxicillin/clavulanic acid alone) or active drugs not affecting the cell wall (rifampicin). Overall, however, the trDNA assay did not correlate well with sputum culture-based decline of viable bacteria. This is possibly due to trDNA reflecting the killing of both culturable and non-culturable bacteria and should be explored further.

**IMPORTANCE**   This study presents the results of the evaluation of a novel method for the detection of *Mycobacterium tuberculosis*, the causative agent of tuberculosis, in urine. Detecting parts of the mycobacteria in urine is of particular interest as it allows us to use a sample that is easy to obtain and that does not require uncomfortable procedures or safety precautions like obtaining sputum for culture, which is the most commonly used sample in the diagnosis of tuberculosis. In certain groups of individuals who cannot produce sputum, for example, children, non-sputum-based methods have particular importance. We found that the method tested was able to detect bacterial killing by active antibiotics that disrupt the cell wall and lead to fragmentation of bacteria. However, the assay can't detect inactive bacteria or bacteria that are active with an intact cell wall.

**KEYWORDS**   *Mycobacterium tuberculosis*, early bactericidal activity, anti-tuberculous treatment

Tuberculosis remains a major global threat; 10.6 million people fell ill and 1.6 million died from tuberculosis in 2021 (1). Diagnosis of pulmonary tuberculosis relies mainly on sputum, but in particular groups, for example, children or people living with HIV, sputum-based methods are less sensitive (2). In these cases, alternative non-invasive methods for rapid and precise tuberculosis diagnosis are urgently needed, as well as more accurate markers for monitoring treatment response. Detection of cell-free transrenal mycobacterial DNA (trDNA) in urine is a promising non-invasive assay that has the potential to improve tuberculosis diagnosis.

Address correspondence to Christoph Lange, clange@fz-borstel.de.

C.L. has received speaker honoraria from Gilead, G.S.K., Insmed, and Janssen, and has been a member of the scientific advisory board of Insmed outside of the scope of this study.

See the funding table on p. 6.

It is believed that small fragments of the *Mycobacterium tuberculosis* (*Mtb*) DNA can penetrate the renal barrier and be detected in the urine (3). In untreated patients, the assay should detect fragments of *Mtb* in the urine due to bacterial killing and cell lysis initiated by the patient's immune response at the site of the disease. The initiation of active treatment, particularly with agents disrupting the cell wall and thereby promoting cell lysis, would be expected to temporarily increase the presence of the fragments, followed by a slow decrease and their eventual disappearance.

A real-time PCR (RT-PCR)-based urine trDNA assay has been developed which detects a 67-base pair fragment of IS6110, a repetitive insertion element highly specific for bacteria from the *Mtb* complex (4, 5). The aim of this study was to assess the performance of the novel trDNA assay for the detection of *Mtb* and comparison to the 2-week anti-tuberculosis therapy response in an early bactericidal activity (EBA) study with six different treatments.

## MATERIALS AND METHODS

The study was conducted on a total of 288 urine samples from 72 HIV-negative participants with newly diagnosed, acid-fast-bacilli sputum-positive rifampicin-susceptible pulmonary tuberculosis enrolled in a 14-day EBA study in Cape Town, South Africa. Treatments included meropenem, ertapenem, or rifampicin in combination with amoxicillin/clavulanate, amoxicillin/clavulanate alone, or standard combination treatment [isoniazid, rifampicin, pyrazinamide, and ethambutol (HRZE)] (Table 1). The study protocol was reviewed and approved by Pharma-Ethics (South Africa, reference no. 170516584); all participants signed informed consent for participation in the study. Urine was collected at baseline and days 3, 7, and 14 of treatment and stored with the addition of ethylenediaminetetraacetic acid (EDTA). Mycobacterial trDNA was isolated with the Wizard Plus Minipreps DNA Purification System (Promega, USA) and detected on a LightCycler 480 RT-PCR platform (Roche, Switzerland). Amplification mix included 1× GoTaq Probe qPCR Master Mix (Promega, USA; cat. no. A6101), 500 nM forward (5′-ACCAGCACCTAACCGGCTGT-3′) and reverse (5′-GTAGGCGAACCCTGCCCAG-3′) primers, and 200 nM probe (6-FAM-5′-TAGCAGACCTCACCTATGTGTCGACCTG-3′-BBQ). Each sample was analyzed in triplicate; a mean cycle threshold (Ct) value was used in the analysis. Data analysis was performed on all available results after their initial quality assessment using the proprietary software. A PCR Ct value of 36 was chosen as a cutoff point. Samples with a Ct value below 36 were considered positive for the presence of *Mtb*, while samples with Ct equal to 36 or higher were considered negative. PCR results were compared with the mean time to positivity (TTP) of two sputum cultures collected on the same day, used as a marker of bacterial load and drug activity. A TTP cutoff of 7 days was used to group cultures with high (TTP < 7 days) and low (TTP ≥ 7 days) bacterial loads. The correlation between a positive PCR result and bacterial load was analyzed using the Fisher exact probability test.

**TABLE 1**  Distribution of positive PCR results (Ct < 36) by study arm over the course of treatment[a]

| Study arm | Positive PCR result (*n/N*, %) | | | |
|---|---|---|---|---|
| | Day 1 | Day 3 | Day 7 | Day 14 |
| Meropenem 6 g once daily/Amx/CA | 4/8 (50.0%) | 5/8 (62.5%) | 5/8 (62.5%) | 4/8 (50.0%) |
| Meropenem 3 g twice daily/Amx/CA | 2/10 (20.0%) | 8/10 (80.0%) | 6/10 (60.0%) | 1/10 (10.0%) |
| Ertapenem IM 1 g once daily/Amx/CA | 4/11 (36.4%) | 3/11 (27.3%) | 7/11 (63.6%) | 4/11 (36.4%) |
| Ertapenem IV 1 g once daily/Amx/CA | 2/11 (18.2%) | 2/11 (18.2%) | 4/11 (36.5%) | 4/11 (36.5%) |
| Amx/CA | 5/16 (31.2%) | 2/16 (12.5%) | 4/16 (25.0%) | 4/16 (25.0%) |
| Rifampicin 35 mg/kg once daily/Amx/CA | 2/11 (18.2%) | 3/11 (27.3%) | 1/11 (9.1%) | 2/11 (18.2%) |
| HRZE standard dose | 1/5 (20.0%) | 3/5 (60.0%) | 4/5 (80.0%) | 1/5 (20.0%) |
| Total | 20/72 (27.8%) | 26/72 (36.1%) | 31/72 (43.0%) | 20/72 (27.8%) |

[a]Amx, amoxicillin; CA, clavulanate; HRZE, standard of care (isoniazid, rifampicin, pyrazinamide, and ethambutol); IM, intramuscularly; IV, intravenously.

## RESULTS

In total, *Mtb* was detected in 97 of the 288 urine samples (33.7%) with the trDNA assay. The highest proportion of positive samples was observed on days 3 and 7 (Fig. 1). This trend was especially notable in the meropenem and HRZE arms and to a lesser extent in the ertapenem arm. The arms with amoxicillin/clavulanate with and without rifampicin had few positive samples (Table 1; Fig. 1).

The proportion of cultures with high bacterial load on TTP decreased steadily from 75.7% on day 1 to 51.6% on day 14 (Table S1; Fig. 2). When analyzed by the study arm, the trend from high load to low bacterial load over time was seen in the rifampicin, meropenem, and HRZE arms. Almost no effect was seen for ertapenem and amoxicillin/clavulanate alone in keeping with the results of the EBA (Table S2; Fig. S1).

A trend of the highest proportion of positive trDNA assay results on day 7 with a decrease by day 14 was observed in participants regardless of bacterial load when analyzing the corresponding TTP and trDNA results (Fig. S2). However, the proportion of positive results was not significantly different between groups with high and low bacterial load and did not show correlation (Table S3).

## DISCUSSION

In this study, we validated a novel trDNA assay on urine from a cohort of participants with pulmonary tuberculosis and positive *Mtb* culture from sputum collected within a 2-week study of various anti-tuberculosis agents.

trDNA was detected in only one-third of the samples, indicating a low sensitivity of the assay. One of the reasons for the low sensitivity may be a low concentration of

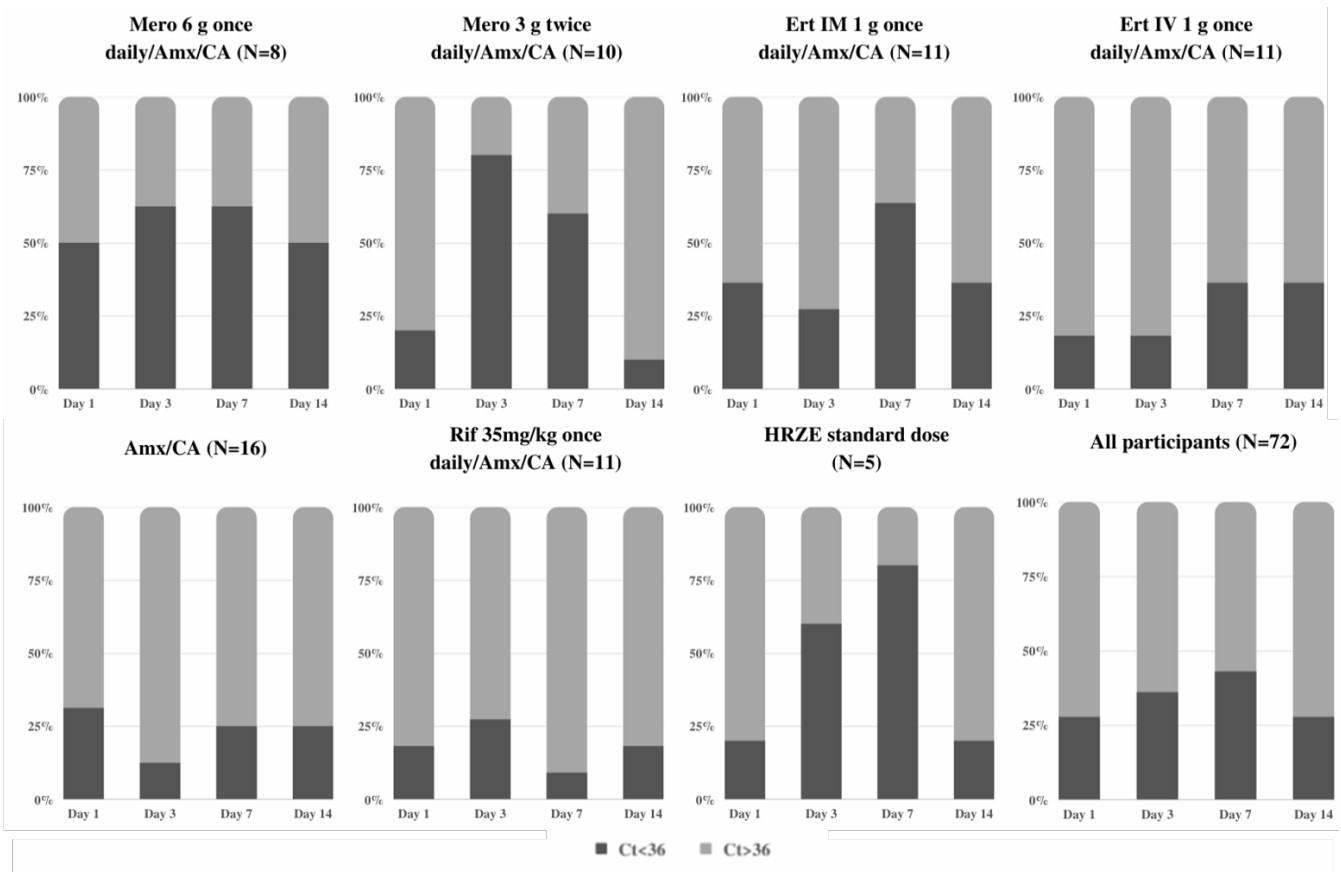

**FIG 1** Distribution of positive PCR results (Ct < 36) by study arms. Mero, meropenem; Ert, ertapenem; Amx, amoxicillin; CA, clavulanate; Rif, rifampicin; IM, intramuscularly; IV, intravenously. The number in the brackets indicates the number of study participants in each arm.

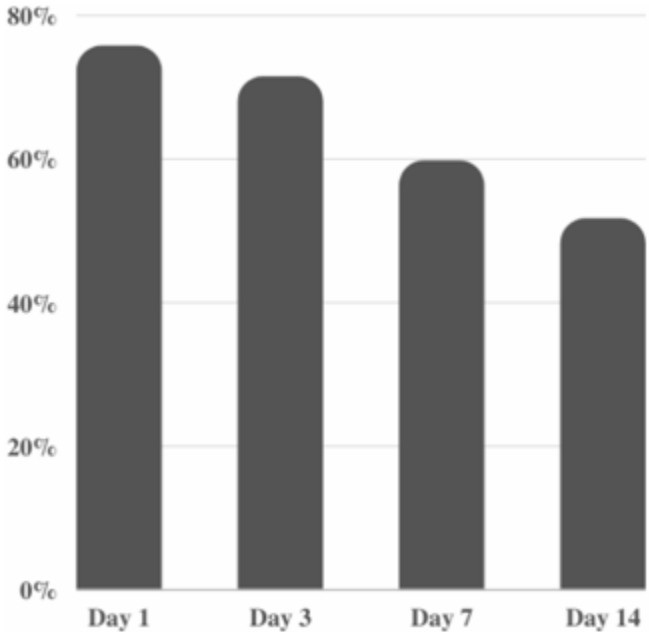

**FIG 2** The proportion of cultures with high (TTP < 7 days) bacterial load over the course of treatment.

mycobacterial DNA in urine. In a previous study, the median urinary *Mtb* cell-free DNA concentration in treatment-naive patients was 6.3 copies/mL; it rose to 25 copies/mL 1 week after treatment initiation but decreased again to 2.5 copies/mL at week 11 (6). In our study, the concentration of DNA in samples was probably too low and reached detectable levels only after several days of effective treatment. Also, the performance of the assay depends on preanalytical and analytical factors. In previous studies, the sensitivity and specificity of trDNA assays ranged widely from 18% to 100%, and the performance of commercial kits for the isolation of trDNA from urine varied widely (7). Further studies comparing the performance of various isolation kits are necessary. The preservation solution can affect the DNA recovery rate: samples preserved in EDTA have a significantly lower positivity rate than those preserved in the Streck reagent (Streck, USA) (8). All samples in our study were preserved in EDTA which might have affected the DNA recovery rate.

To evaluate the reproducibility of the assay, we ran every DNA sample in triplicate. The Ct values from three reactions were usually similar which points to the good reproducibility of the assay. Also, it should be noted that in most of the cases, if the sample collected on day 1 was PCR positive, samples collected on later time points would also be positive. Interestingly, in the majority of these cases, two sputum samples collected on the same day were also positive with a TTP of less than 7 days which points at the association between higher bacterial load and the positive PCR result (Table S4).

There was a trend in the proportion of positive results over the duration of treatment with the highest positive proportion observed on day 7, declining towards day 14. This trend was observed in meropenem and HRZE arms, less so for ertapenem, but not in amoxicillin/clavulanate or rifampicin arms. This may be explained by the effective killing of the bacteria with cell wall fragmentation over the first week of treatment and the filtering of Mycobacterial DNA through the kidney into the urine. By day 14, fewer bacteria are killed, and therefore, less DNA is detected. That this trend was seen specifically with antibiotics that have activity against the cell wall, namely, meropenem and isoniazid and ethambutol in the control arm, suggests that the assay is detecting fragmentation products specific to the cell wall. Rifampicin is an RNA synthesis inhibitor with excellent EBA where one would expect to see fragmentation products, suggesting that the drug mechanism of action plays a role in the assay performance. This supports the conclusion that detection of trDNA depends on (i) a treatment being active as

reflected by sputum culture and (ii) the treatment affecting cell wall integrity leading to cell lysis and fragmentation.

Further analysis demonstrated that a positive trDNA result did not correlate well with a higher bacterial load on culture. We know, however, that only a fraction of bacteria in sputum is culturable. These discrepancies could thus be explained by trDNA measuring killing in a much wider range of bacteria than can be monitored by sputum culture. There may be a role for the trDNA assay to provide supplementary evidence for bactericidal activity of experimental combinations.

Our study has a number of limitations. Firstly, due to low sensitivity of the assay, we were unable to explore variations in cycle threshold positivity across treatment regimens. Secondly, all samples were preserved in EDTA, which likely resulted in a reduced DNA recovery rate. We did not conduct a comparison of the sensitivity of the assay under different conditions, such as utilizing samples preserved in different solutions or extracted using various DNA isolation kits.

In summary, the trDNA assay in its current form has insufficient sensitivity for the diagnosis of TB and cannot replace standard microbiological methods for the evaluation of treatment response in 2 weeks EBA studies at this point in time. However, we believe that the temporal association of trDNA detection with the expected maximum killing by active agents on the bacterial cell wall, but not inactive agents or active agents not affecting the cell wall, is proof-of-concept that trDNA can be developed into a useful biomarker.

Studies evaluating the performance of other non-sputum-based methods of detection of cell-free *Mtb* DNA in EBA studies are still urgently needed. For example, an ultrasensitive clustered regularly interspaced short palindromic repeats (CRISPR)-Cas12a-powered fluorescence assay to detect *M. tuberculosis* cell-free DNA in blood has recently been shown to have a high sensitivity and specificity, even in a pediatric cohort (9). However, its feasibility for monitoring of treatment response in an EBA study remains unknown.

Non-sputum-based assays show great promise as diagnostic and treatment monitoring tools for tuberculosis, particularly in patients where obtaining a sputum sample is challenging such as children or people living with HIV.

## ACKNOWLEDGMENTS

This work was supported by Horizon 2020 grant no. 733079 (AnTBiotic: progressing TB drug candidates to clinical proof of concept). The funders had no role in study design, data collection and interpretation, or the decision to submit the work for publication.

We thank all study participants and their families, nurses, physicians, laboratory technicians, and study personnel involved in this study.

I.K., J.H., C.U., V.D.J., A.D., and C.L. made substantial contributions to the conception and design of this study, the enrollment of study participants, and the acquisition of samples. F.K., D.H., T.G., and J.H. designed the novel assay and tested samples. I.K. analyzed the data and wrote the manuscript. All authors critically revised the manuscript for important intellectual content, gave final approval of the version to be published, and agreed to be accountable for all aspects of this work.

## AUTHOR AFFILIATIONS

[1]Division of Clinical Infectious Diseases, Research Center Borstel, Leibniz Lung Center, Borstel, Germany

[2]German Center for Infection Research, Borstel, Germany

[3]Respiratory Medicine and International Health, University of Lübeck, Lübeck, Germany

[4]Department of Infectious Disease, Faculty of Medicine, Imperial College London, London, United Kingdom

[5]Department of Internal Medicine I, University Medical Center Schleswig-Holstein, Kiel, Germany

[6]National Reference Center, Research Center Borstel, Borstel, Germany
[7]Division of Histology, Research Center Borstel, Leibniz Lung Center, Borstel, Germany
[8]Airway Research Center North, Member of the German Center for Lung Research, Großhansdorf, Germany
[9]TASK Applied Science, Cape Town, South Africa
[10]Baylor College of Medicine, Houston, Texas, USA

## AUTHOR ORCIDs

Irina Kontsevaya http://orcid.org/0000-0003-3421-7550
Caryn M. Upton http://orcid.org/0000-0002-7503-3320

## FUNDING

| Funder | Grant(s) | Author(s) |
|---|---|---|
| EC \| Horizon 2020 Framework Programme (H2020) | 733079 | Irina Kontsevaya |
| | | Jan Heyckendorf |
| | | Frauke Koops |
| | | Caryn M. Upton |
| | | Veronique De Jager |
| | | Andreas Diacon |
| | | Christoph Lange |

## AUTHOR CONTRIBUTIONS

Irina Kontsevaya, Conceptualization, Formal analysis, Investigation, Methodology, Project administration, Supervision, Visualization, Writing – original draft, Writing – review and editing | Jan Heyckendorf, Conceptualization, Data curation, Funding acquisition, Methodology, Project administration, Supervision, Validation, Writing – review and editing | Frauke Koops, Data curation, Investigation, Methodology, Writing – review and editing | Doris Hillemann, Investigation, Methodology, Writing – review and editing | Torsten Goldmann, Methodology, Writing – review and editing | Caryn M. Upton, Conceptualization, Funding acquisition, Writing – review and editing | Veronique De Jager, Conceptualization, Funding acquisition, Writing – review and editing | Andreas Diacon, Conceptualization, Investigation, Resources, Writing – review and editing | Christoph Lange, Conceptualization, Project administration, Resources, Supervision, Writing – review and editing

## DATA AVAILABILITY

The data that support the findings of this study are available from the corresponding author upon reasonable request.

## ADDITIONAL FILES

The following material is available online.

### Supplemental Material

**Supplemental file 1 (Spectrum02348-23-S0001.doc).** Fig. S1 and S2 and Tables S1 to S4

### Open Peer Review

**PEER REVIEW HISTORY (review-history.pdf).** An accounting of the reviewer comments and feedback.

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
