## [Reviewer comments · Microbiology Spectrum]

Microbiology Spectrum

Transrenal *Mycobacterium tuberculosis* DNA in pulmonary tuberculosis patients during the first 14 days of treatment

Irina Kontsevaya, Jan Heyckendorf, Frauke Koops, Doris Hillemann, Torsten Goldmann, Caryn Upton, Veronique De Jager, Andreas Diacon, and Christoph Lange

Corresponding Author(s): Irina Kontsevaya, Forschungszentrum Borstel Leibniz Lungenzentrum

Review Timeline:

Submission Date:	June 5, 2023
Editorial Decision:	July 17, 2023
Revision Received:	August 17, 2023
Accepted:	September 11, 2023

Editor: Po-Yu Liu

Reviewer(s): Disclosure of reviewer identity is with reference to reviewer comments included in decision letter(s). The following individuals involved in review of your submission have agreed to reveal their identity: Sangeeta Tiwari (Reviewer #1)

Transaction Report:

DOI: <https://doi.org/10.1128/spectrum.02348-23>

July 17, 2023

Dr. Irina Kontsevaya
Forschungszentrum Borstel Leibniz Lungenzentrum
Clinical Infectious Diseases
Parkallee 1-40
Borstel 23845
Germany

Re: Spectrum02348-23 (Transrenal *Mycobacterium tuberculosis* DNA in pulmonary tuberculosis patients during the first 14 days of treatment)

Dear Dr. Irina Kontsevaya:

Link Not Available

Sincerely,

Po-Yu Liu

Journals Department
Reviewer comments:

Reviewer #1 (Comments for the Author):

The manuscript of Kontsevaya et al. addresses the importance of biological tool, transrenal *Mycobacterium tuberculosis* DNA detection in urine sample of patients through RT-PCR. The authors demonstrate that this non-invasive technique could be good marker for detection of Mtb after early TB drugs treatment. Authors also mentioned that this method is not well correlated with sputum-based method. The manuscript is interesting, and presents a new methods for detection of Mtb.

I have some major and minor comments on the manuscript:

Major comment:

1. Sensitivity of this technique is attributed to action of cell wall targeting antibiotics. Will be interesting to see how well this technique works with urine samples from patients undergoing isoniazid (first line TB drug) and other cell wall targeting antibiotics

therapy?

2. Is this method applicable for detection of bacterial load if sample is lated at late time points as after 2 weeks?

3. In this manuscript authors did not discuss Mtb severity levels of the patients.

Minor comment:

Line 89: Is total number of samples 288 or 72 used in study?

Reviewer #2 (Comments for the Author):

In this study the authors applied to detection of trDNA to urine samples to assess the potential of trDNA to detect bacterial killing in an EBA study. Although trDNA detection is too insensitive to be used as a diagnostic its use to follow bacterial killing in EBA studies would be interesting. The authors do indeed observe a moderate increase on the proportion of positive samples during treatment (Ct<36) but this peak does not correlate with the peak in EBA observed by culture. The authors suggest this may be due to killing of a different bacterial population which is indeed a possibility This is an interesting study and deserves to be reported.

I have a couple of comments questions:

What about the actual Ct values when was the lowest Ct value observed? In lines 168-68 the authors state they were not able to compare Ct values. How consistent were the Ct values between individual patients, what was the range of Ct values obtained for all patinets?

Would it be possible to normalise to the day 1 Ct value (if positive) of patients with positive results? Or were the positive results randomly distributed between the patients, i.e. many / most patients only positive at one time point? Or to rephrase how many patients positive at time point one were positive/not positive at later time points? Would it be possible at least to present this information i.e. the samples positive at each time point (and the Ct value) for each patient may be in a supplementary file.

Staff Comments:

Preparing Revision Guidelines

Please return the manuscript within 60 days; if you cannot complete the modification within this time period, please contact me. If you do not wish to modify the manuscript and prefer to submit it to another journal, please notify me of your decision immediately so that the manuscript may be formally withdrawn from consideration by Microbiology Spectrum.

The manuscript of Kontsevaya et al. addresses the importance of biological tool, transrenal *Mycobacterium tuberculosis* DNA detection in urine sample of patients through RT-PCR. The authors demonstrate that this non-invasive technique could be good marker for detection of *Mtb* after early TB drugs treatment. Authors also mentioned that this method is not well correlated with sputum-based method. The manuscript is interesting, and presents a new methods for detection of *Mtb*. I have some major and minor comments on the manuscript:

Major comment:

1. Sensitivity of this technique is attributed to action of cell wall targeting antibiotics. Will be interesting to see how well this technique works with urine samples from patients undergoing isoniazid (first line TB drug) and other cell wall targeting antibiotics therapy?
2. Is this method applicable for detection of bacterial load if sample is lated at late time points as after 2 weeks?
3. In this manuscript authors did not discuss *Mtb* severity levels of the patients.

Minor comment:

Line 89: Is total number of samples 288 or 72 used in study?

Manuscript reference number: Spectrum02348-23

Dr. Po-Yu Liu

Editor

Microbiology Spectrum

Dear Dr. Liu,

We would like to express our gratitude to you and the peer reviewers for the evaluation of our submission entitled

Transrenal Mycobacterium tuberculosis DNA in pulmonary tuberculosis patients during the first 14 days of treatment

Following the helpful comments provided we have revised the manuscript accordingly. Please find a point-by-point response to the comments below.

Reviewers' comments

Reviewer #1

Comment #1 by Reviewer #1:

The manuscript of Kontsevaya et al. addresses the importance of biological tool, transrenal Mycobacterium tuberculosis DNA detection in urine sample of patients through RT-PCR. The authors demonstrate that this non-invasive technique could be good marker for detection of Mtb after early TB drugs treatment. Authors also mentioned that this method is not well correlated with sputum-based method. The manuscript is interesting, and presents a new methods for detection of Mtb.

I have some major and minor comments on the manuscript:

Authors' response to the comment #1 of Reviewer #1:

We thank the Reviewer for the high evaluation of our manuscript.

Comment #2 by Reviewer #2:

Major comment:

Sensitivity of this technique is attributed to action of cell wall targeting antibiotics. Will be interesting to see how well this technique works with urine samples from patients undergoing isoniazid (first line TB drug) and other cell wall targeting antibiotics therapy?

Authors' response to comment #2 of Reviewer #2:

We thank the Reviewer for the comment. Indeed, in the standard combination treatment containing isoniazid, the assay detected a higher proportion of PCR positive results than in those containing drugs not targeting the cell wall, for example, amoxicillin/clavulanate arm (80% vs 25% on Day 7, Table 1). Based on this observation, we hypothesise that the detection of trDNA depends on whether the treatment affects cell wall integrity and leads to cell lysis and fragmentation. This hypothesis would need to be evaluated in future research which is beyond the scope of the reported work.

Comment #3 by Reviewer #2:

Is this method applicable for detection of bacterial load if sample is lated at late time points as after 2 weeks?

Authors' response to comment #3 of Reviewer #2:

We thank the Reviewer for the comment. There are no data yet for the assay described in this study for samples collected after 2 weeks of treatment. Overall, data on the use of detection of Mycobacterial trDNA in urine for monitoring of treatment response are very limited as most of the studies so far have focused on using the assay for the diagnosis of TB. However, in a study by Labugger et al., Infection 2017 (PMID 27798774) a similar qPCR-based assay for the detection of trDNA was used on samples from a cohort of 11 treatment-naïve patients and 11 patients on treatment with weekly sampling during treatment. It was shown that after treatment initiation, trDNA showed a significant reduction in concentration over time reaching undetectable trDNA values at week 12 in 82% of treatment-naïve patients. In our study, the sensitivity of the assay was quite low even within the first 2 weeks of treatment when the bacterial load is at its highest. We therefore assume that after 2 weeks the detection rate would be very low. Further optimisation of the assay and validation on a long-term cohort of TB-infected individuals undergoing longer treatment would be needed.

Comment #4 by Reviewer #2:

In this manuscript authors did not discuss Mtb severity levels of the patients.

Authors' response to comment #4 of Reviewer #2:

We agree with the Reviewer that this aspect has not been discussed in the manuscript. The study participants were enrolled in an Early Bactericidal Activity study of drugs and combination with a strict set of inclusion and exclusion criteria. Participants with newly diagnosed, rifampicin susceptible pulmonary TB with at least 1+ smear microscopy, without HIV co-infection, diabetes, history of TB or signs of extrathoracic TB, among other criteria, were included. These participants are informally considered "healthy" TB patients and in terms of clinical well-being are quite similar. However, burden of disease can vary substantially from 1+ to 3+ and approximately 75% of the participants had cavitory lung disease which will be reported in the main paper currently in draft. trDNA positivity rate was compared to the time to positivity (TTP) of two sputum samples taken on the same day which is the indicator of bacterial load detected at a particular time point.

Comment #5 by Reviewer #2:

Minor comment:

Line 89: Is total number of samples 288 or 72 used in study?

Authors' response to comment #5 of Reviewer #2:

The study was conducted on 288 samples collected from 72 study participants. Four urine samples were collected from each participant during the 2-week treatment: on Day 1, 3, 7, and 14. This resulted in 288 urine samples in total.

Reviewer #2

General comment of Reviewer #2:

In this study the authors applied to detection of trDNA to urine samples to assess the potential of trDNA to detect bacterial killing in an EBA study. Although trDNA detection is too insensitive to be used as a diagnostic its use to follow bacterial killing in EBA studies would be interesting. The authors do indeed observe a moderate increase on

the proportion of positive samples during treatment (Ct<36) but this peak does not correlate with the peak in EBA observed by culture. The authors suggest this may be due to killing of a different bacterial population which is indeed a possibility This is an interesting study and deserves to be reported.

Authors' response to the general comment of Reviewer #2:

We thank the Reviewer for the high evaluation of our work.

Comment #1 by Reviewer #2:

I have a couple of comments questions:

What about the actual Ct values when was the lowest Ct value observed? In lines 168-68 the authors state they were not able to compare Ct values. How consistent were the Ct values between individual patients, what was the range of Ct values obtained for all patients?

Authors' response to comment #1 of Reviewer #2:

We thank the Reviewer for this interesting comment. In the study, Ct values ranged from 31 to 40 (the upper limit of measurement of the qPCR system used in the study). Each sample was tested in triplicate and the Ct values from three reactions were usually similar which points to the reproducibility of the method. It was not possible to analyse the consistency of Ct values between individual patients as the positivity rate was low but it seems that the results were not very consistent. Further optimisation of the assay is needed to improve the sensitivity and be able to conduct more a detailed analysis.

Comment #2 by Reviewer #2:

Would it be possible to normalise to the day 1 Ct value (if positive) of patients with positive results? Or were the positive results randomly distributed between the patients, i.e. many / most patients only positive at one time point? Or to rephrase how many patients positive at time point one were positive/not positive at later time points? Would it be possible at least to present this information i.e. the samples positive at each time point (and the Ct value) for each patient may be in a supplementary file.

Authors' response to comment #2 of Reviewer #2:

We thank the Reviewer for this suggestion. The table with all results would be too long so we added Table S4 that only includes 20 study participants who had a positive PCR

result on day 1. It can be seen that in most of the cases, if the sample collected on day 1 was PCR-positive samples on later time points will also be positive. Interestingly, in the majority of these cases, two sputum samples collected on the same day are also positive with a time to positivity of less than 7 days. We added these observations in Lines 153-159.

We thank the Editor and the peer reviewers for their support in improving the manuscript. We hope that we have addressed most of the comments adequately and the manuscript can be accepted for publication in Microbiology Spectrum.

On behalf of all co-authors,

Irina Kontsevaya PhD

September 11, 2023

Dr. Irina Kontsevaya
Forschungszentrum Borstel Leibniz Lungenzentrum
Clinical Infectious Diseases
Parkallee 1-40
Borstel 23845
Germany

Re: Spectrum02348-23R1 (Transrenal *Mycobacterium tuberculosis* DNA in pulmonary tuberculosis patients during the first 14 days of treatment)

Dear Dr. Irina Kontsevaya:

Your manuscript has been accepted, and I am forwarding it to the ASM Journals Department for publication. You will be notified when your proofs are ready to be viewed.

Sincerely,

Po-Yu Liu
Editor, Microbiology Spectrum
